# Safety and Effectiveness of Percutaneous Endoscopic Gastrostomy May Be Improved by Proper Pre- and Post-Positioning Management of Elderly Patients with Multimorbidity

**DOI:** 10.3390/nu16172893

**Published:** 2024-08-29

**Authors:** Paolo Orlandoni, Nikolina Jukic Peladic

**Affiliations:** 1Clinical Nutrition Unit, National Institute of Health and Science on Aging, IRCCS INRCA Ancona, Via della Montagnola 81, 60127 Ancona, Italy; p.orlandoni@inrca.it; 2Vivisol Srl. at Clinical Nutrition Unit, National Institute of Health and Science on Aging, IRCCS INRCA Ancona, Via della Montagnola 81, 60127 Ancona, Italy

**Keywords:** percutaneous endoscopic gastrostomy, complications, mortality, geriatric patients, home enteral nutrition

## Abstract

Introduction: The main risk factors for major complications and early mortality after the positioning of percutaneous endoscopic gastrostomy (PEG) reported in the literature are old age, multimorbidity, and the use of inappropriate methods for PEG positioning. A proper PEG positioning technique and adequate post-positioning patient management and surveillance are the main protective factors, but the information on protective factors in the literature is much poorer. The aim of this study was to provide more information on PEG-related complications and mortality in geriatric patients treated with long-term enteral nutrition administered by PEG according to a specific home enteral nutrition (HEN) protocol. Methods: This was a retrospective study based on data from 136 elderly patients in whom PEG was positioned from 2017 to 2023 at the geriatric hospital IRCCS INRCA, Ancona (Italy), 88 of whom were treated with HEN. Data on PEG-related complications, duration of HEN, hospitalizations, and mortality were analyzed. Results: No complications were registered during or immediately after the PEG positioning. The prevalence of a major complication—buried bumper—was in the lower limit of the range reported in the literature (4.32%). The prevalence of minor complications such as peristomal leakage, inadvertent tube removal, and granulation tissue was higher than that reported in the literature (14.71%, 23.53%, 29.41%), while tube blockage and peristomal site infection were less frequent (8.82%, 38.23%). Three hospitalizations for PEG-related complications occurred. Both the all-cause 30-day mortality and within-two-months mortality were lower than those in the literature (1.92% and 3.84%). Conclusions: The impact of the risk factors recognized by the literature on complications and mortality could be probably mitigated by improving the PEG placement techniques and pre- and post-PEG placement patient management practices. Data on the prevalence of complications and mortality must be interpreted in correlation to this information.

## 1. Introduction

Enteral nutrition (EN) or, as it is also called, tube feeding (TB), is a treatment that consists of administering macro- and micronutrients contained in commercial feeding formulas directly into the stomach of subjects that cannot meet their nutritional needs safely by oral feeding [1,2]. Age-related pathologies and conditions—dementia and other neurological diseases, sarcopenia, difficulty swallowing (dysphagia), and polypharmacy—often cause malnutrition or the impossibility of eating safely by mouth, meaning that EN is particularly frequent among elderly patients [3].

Scientific societies provide evidence-based recommendations for the provision of EN [4,5,6,7]. The evidence on percutaneous endoscopic gastrostomy (PEG), which is the most-used access route for the long-term administration of EN, suggests that the main risk factors for major complications and early mortality after PEG positioning are old age, multimorbidity, and the use of inappropriate methods for PEG positioning [8,9]. On the contrary, a proper PEG positioning technique and adequate post-positioning patient management and surveillance have been identified as the main protective factors [10,11]. Yet, the evidence on protective factors is very scarce or non-existent, as is the case for information on patients’ follow-up after PEG placement. Even the data on the prevalence of PEG complications are confounding since they concern patients of different ages; vary consistently, going from 16% up to 70%; and are mostly provided without relating them to the overall duration of therapy [12,13,14,15].

Assuming that the adoption of and compliance with rigorous protocols for the management of patients pre and post PEG placement can positively affect the safety and effectiveness of EN via PEG in older patients and reduce the complications and mortality, we performed a retrospective analysis of data from a prospective observational study to describe the results achieved in a population of elderly patients in whom PEG tubes were placed at the Scientific Institute for Research, Hospitalization and Healthcare (IRCCS), National Institute of Health and Science on Aging (INRCA), Ancona, Italy, following a specific protocol, and who were followed after discharge according to the protocol of the INRCA Regional Center for Home Artificial Nutrition (HAN). 

## 2. Materials and Methods

### 2.1. Study Design

This study is based on data from 136 elderly patients in whom PEGs were positioned from 2017 to 2023, 88 of whom were treated with home enteral nutrition (HEN). Data were collected as part of the project “Observational study of the elderly patients treated with Home Enteral Nutrition (HEN)”, which was approved by the Ethics Committee of the IRCCS- INRCA Ancona (Italy) in compliance with Italian national rules and standards for ethical research conduct (Identification Code CdB SC/14/442—Determination n. 16 from 10 September 2014). The project was designed as an observational, prospective, cohort, “umbrella” study to collect evidence on different issues and problems concerning elderly patients treated with HEN. All details about the project and protocols adopted for the provision of HEN have been previously published [16,17,18].

The study question that was formulated using the PICO statement was whether the adoption and compliance with rigorous protocols for the selection of the elderly patients suitable for PEG placement and for home management, such as those adopted at INRCA, aid in attaining good results in terms of PEG-related complications and mortality in elderly patients [19,20]. 

The primary outcome of this retrospective study was the prevalence of different complications after PEG placement. Major complications that are systemic and life-threatening included aspiration pneumonia, hemorrhage, buried bumper syndrome, perforation of the bowel, and necrotizing fasciitis. Minor complications included PEG-tube occlusion, deterioration, breakage, dislocation, peritonitis, granulation tissue, redness, and enlargement of the stoma site. The secondary outcome was all-cause mortality at 30 days and within-two-months mortality.

#### PEG Positioning and Follow-Up of Patients with PEG in HEN

PEGs were positioned at IRCCS—INRCA hospital in patients after the indication of the physician from Clinical Nutrition and after the patients or their administrators gave written informed consent. The protocol details are shown in Table 1. A multi-professional team formed by anesthetists, endoscopists, and nutritionists participated in PEG positioning. 

Detailed demographic (age, gender, residence) and clinical data (main pathology, comorbidities, drug therapies, nutritional status, indications for PEG positioning) were collected from the patient’s documentation before the PEG positioning. All data were recorded and managed within a dedicated database at the Clinical Nutrition Unit of IRCCS INRCA, Ancona.

The patients were discharged from the hospital only when their clinical conditions were stable and if there were no early complications after PEG placement. The PEG was replaced after 10–12 months with a balloon gastrostomy tube. For patients enrolled in the INRCA HAN service, home visiting staff formed by nurses visited patients once a month, directly at their homes or nursing homes. Home visiting staff were in contact with hospital staff formed by nutritionists, dietitians, and speech therapists. According to the INRCA protocol, during the home visits, nurses assess patients’ nutritional status and overall clinical conditions and check for HEN-related complications: tube-related, gastrointestinal, and metabolic. When possible, HEN-related complications are resolved at home, during the home visit. Otherwise, priority access to the hospital for out-hospital visits or hospitalization is organized. During each monthly home visit, the nursing staff member checks the correct positioning of the external and internal bumper of the PEG tube and ensures that the external bumper is fixed; without pressing, they check for any signs of redness/peritonitis; and, if necessary, they carry out the medication of the stoma. A PEG change is provided every 4 months. When the nurses responsible for home visits judge that an immediate change in PEG is necessary, subjected to authorization from the Clinical Nutrition physician, the PEG is changed immediately and directly at the patient’s home. All data on clinical conditions and nutritional status, which are assessed during the home visit, as well as the information on complications and hospitalizations during the month, are recorded and managed within a dedicated database that may be consulted by the hospital staff in real time. For this study, data on PEG-related complications, the overall duration of HEN therapy after the PEG positioning, hospitalizations, and mortality for PEG-related complications were analyzed.

### 2.2. Data Analyses

The collected data were coded, processed, and analyzed using SPSS (Statistical Package for the Social Sciences) version 22 for Windows^®^ (IBM SPSS Inc., Chicago, IL, USA). Descriptive statistics was used to analyze the clinical characteristics of the patients, complications, and mortality. The normality of the continuous variables was tested using the Shapiro–Wilk test, and variables were reported as mean and standard deviation (SD), or median and interquartile range (IQR), based on their distribution. Categorical variables were expressed as absolute and relative frequencies. 

## 3. Results

From 2017 to 2023, a PEG tube was positioned successfully in 134 of 136 geriatric patients at IRCCS INRCA hospital (98.48%). The characteristics of the patients are presented in the first column of Table 2. The mean age of patients was 81.34 ± 9.19 years. Patients mostly had neurological diseases. Almost 80% of patients were multimorbid, and 66.67% had three or more diseases. More than a half of subjects was undernourished (Body Mass Index (BMI) < 22 kg/m^2^). The main indication for PEG positioning was severe dysphagia (46.43%) and anorexia.

The median duration of hospitalization after the PEG positioning was 4 days (min 1, max 70).

Two PEG placements were impossible because of the impracticality of the transillumination of the gastric wall. No complications were registered during or immediately after PEG positioning.

After PEG positioning, 88 patients (67.69%) were followed up by INRCA’s Home Enteral Nutrition Service and were receiving regular monthly home visits; 52 patients (59.09%) were residing at home and 36 (40.91%) in nursing homes. As it is shown in Table 2, the characteristics of the patients followed-up by INRCA’s HEN service differed for gender (63.36% F in HEN vs. 54.41% overall) and for the prevalence of dementia, undernourishment, and pressure ulcers, which were all higher in subjects treated with HEN (61.36% vs. 54.15%; 47.72% vs. 55.00%; 45.45% vs. 36.84% respectively). These differences were statistically significant (*p* < 0.05).

Overall, during the 49.752 days of HEN provided, 394 HEN-related complications were registered, corresponding to 0.0079 complications/day of HEN therapy.

As it is shown in Table 3, the PEG-related complications were highly prevalent (70.56%) among different HEN-related complications (PEG-related, gastrointestinal, and metabolic). On the contrary, metabolic complications only made up 1.01%. Fifty-four patients (61.37%) who faced PEG-related complications registered a median number of three complications during HEN therapy (min 1; max 30). The number of PEG-related complications was correlated to the duration of the HEN therapy (r = 0.60). More than 27% of patients (24) did not register any HEN-related complications, though they were treated with HEN for a median period of 371 days (min 59; max 701).

Different types of major and minor PEG-related complications are presented in Table 4.

Only one major complication—buried bumper syndrome—occurred, representing 4.32% of the total complications. Granulation tissue and peritonitis represented around 30% of the total complications each. Peristomal site infection was registered in 38.23% of patients, granulation tissue in 29.41%, and dislocation in 29.41%. The number of patients that registered PEG tube deterioration (26.47%) was also quite high.

Almost all complications were solved at home (98.92%), without moving the patient to hospital. Only three (3) hospitalizations for PEG-related complications occurred. They were due to buried bumper syndrome (1) and aspiration pneumonia (1), and one hospitalization was organized for the change in the nutrition therapy. All hospitalizations ended with discharge.

HEN therapy ended in 59.09% (52 pts.) of cases because of death. Two patients (4.54%) turned to oral feeding, while 34.09% (thirty) of subjects were still treated with HEN at the moment of the study. One subject (2.27%) changed the HEN center and was lost to follow-up.

The all-cause 30-day mortality was 1.92%. Within-two-months mortality was 3.84%. The median duration of HEN therapy in patients who ended HEN therapy because of death was 382 days (min 27; max 2290). An increase in the median duration of HEN therapy was registered for the patients who were still treated with HEN at the moment of the study. No 30-day or within-two-months mortality was registered among them.

## 4. Discussion

In this retrospective, observational study, we analyzed the frequency of different PEG-related complications and mortality in geriatric patients in whom PEG was positioned at IRCCS INRCA hospital in the period 2017–2023 and who were treated with home enteral nutrition (HEN) after hospital discharge. No major or minor complications were registered during or immediately after the PEG positioning. The median duration of HEN was 382 days (min 27, max 2290), during which, 394 HEN-related complications were registered. PEG-related complications represented 70.56% of overall complications. However, 36.36% of patients never registered any PEG-related complications. The number of PEG-related complications was correlated to the duration of the HEN therapy (r = 0.60). The only major complication was a buried bumper. Three hospitalizations for PEG-related complications occurred, all ending with hospital discharge. The all-cause 30-day mortality was 1.92%, and the within-two-months mortality was 3.84%.

Our results on the complications and mortality only partially confirm the results of previous studies that identified old age as one of the main risk factors for complications [22].

The literature suggests that old age is the main risk factor for PEG procedure-related complications; pneumoperitoneum, colon injury, gastro-colo-cutaneous fistula, small bowel injury, liver injury, spleen injury, intraperitoneal and retroperitoneal bleeding, and abdominal wall bleeding. In our study, despite the very old age of patients, none of these complications occurred [12,14,23,24,25,26,27]. Although analyses carried out in this study do not allow us to provide clear answers on the reasons for this result, presumably, it could be at least partially attributed to the correct technique used for PEG positioning and to the adequate definition of the conditions that patients needed to have in order to become candidates for PEG positioning [9,15,28].

Comparison of our results on the prevalence of major and minor complications during follow-up with literature data is difficult. Our results were collected, in fact, among patients treated with long-term HEN who were followed uniformly according to the INRCA HEN service protocol after the hospital discharge. On the contrary, results reported in the literature are from studies where the duration of follow-up is rarely specified, and even less information is available on patient surveillance models after PEG positioning. The only major complication registered in our study was buried bumper syndrome. It occurred in 5.88% of patients with PEG, remaining close to the minimum values of the range that, in the literature, ranges from 4.8% to 8.8% [29,30,31,32]. In relation to minor complications, we registered worse results than those reported in the literature for peristomal leakage, inadvertent tube removal, and granulation tissue that were registered in 14.71%, 23.53%, and 29.41% of patients vs. 1–10%, 1.6–4.4%, and 8.4%, respectively, reported in the literature. On the contrary, our results were better with reference to the complications due to tube blockage—occlusion, clogging, or blockage—(8.82% vs. 25–35% in the literature) and peristomal site infection (38.23% vs. 4–50% in the literature) [2,23,33,34,35,36,37,38,39,40,41,42].

The main difference between our results and those from the literature concerns the mortality after PEG positioning. In their review from 2023, Lima et al. reported data from different studies on all-cause 30-day mortality after PEG positioning which was found to reach up to 13.20% in neurologic patients, 23.50% in patients with dementia, 12.70% in patients with cancer, and 22.00% in those with other pathologies [12,43,44,45]. In our study, the all-cause 30-day mortality was only 1.92%. In the same review, the risk factors for early mortality after PEG insertion were exposed, many of which were also registered in our patients. They comprise dementia, urinary tract infection, previous aspiration, diabetes, hypalbuminemia, acute illness, hospitalization, bedsores, higher age, nil-by-mouth, poor nutritional state, low BMI, and the number of comorbidities [12]. Over 50% (54.14%) of the subjects in our study had some form of dementia, 13.43% had DM, 6.82% had albumin levels lower than 2.5 g/dL 10.71% had experienced aspiration episodes before PEG placement, 550.50% had a BMI lower than 21 kg/m^2^, and almost 79.41% of subjects had comorbidities. Still, while in the literature, within-two-months mortality ranges from 9.8% to 15.34%, in our patients, its prevalence was 3.84% [8,45].

This study describes factors that, in the literature, were defined as protective factors for complications and mortality in patients with PEG and how they are managed within the INRCA HEN organizational model.

The present study has some limitations. First of all, it is a descriptive study. Our results in terms of complications and mortality were achieved in a population of elderly patients for whom the same specific protocols were followed both before and after PEG positioning. Since no comparison was made with a control group, it was impossible to determine the effective role of such protocols in improving outcomes. While our study certainly highlights the opportunity to address such analyses in the future, the results that we achieved and presented in this manuscript cannot be used to establish causality and associations. Future studies on complications and mortality in patients with PEG should provide a detailed description of methods used to manage patients in hospital and at home so that the best practices that positively correlate with patients’ safety and quality of life may be identified. Furthermore, future studies should overcome another limitation of the present study, which is that they should include a greater number of patients than we did in ours.

## 5. Conclusions

Complications and early mortality could probably be mitigated even in elderly patients with multimorbidity by applying rigorous protocols that specify correct pre- and post-PEG positioning techniques and patient management practices. Data on the prevalence of complications and mortality after PEG placement must be interpreted in correlation to that information. New studies are needed to provide evidence on the topic in question.

## Figures and Tables

**Table 1 nutrients-16-02893-t001:** Protocol for PEG positioning, Scientific Institute for Research, Hospitalization and Healthcare (IRCCS), National Institute of Health and Science on Aging (INRCA), Ancona, Italy.

Before the PEG positioning	The assessment of indications and contra-indications for PEG positioning	Indications: life expectancy greater than 30 days, normal gastrointestinal function, previous administration of enteral feeding by a nasogastric tube for at least 30 days, written informed consent by the patients or legal administrators Contra-indications: deep metabolic changes, organ failure, ascites, severe and uncorrectable coagulopathy, sepsis, inflammation of the gastric or abdominal wall, conditions that may alter abdominal trans illumination (obesity, previous laparotomies) [21]
	A day before PEG positioning	Suspension of any anticoagulant therapy Enema administration the night before PEG placement Fasting from the night before PEG placement Hair removal from the epigastric area Anesthesia visit ECG report Blood count and coagulation (data relating to a period not exceeding 48 h) Antibiotic prophylaxis one hour before the PEG placement Prompt pump inhibitor one hour before the PEG positioning
PEG positioning	Perioperative actions	Cleaning of the oral cavity Skin disinfection and asepsis of the abdominal insertion site Sedation
	PEG placement	Pull technique with 20 Fr tubes
After the PEG positioning	Post placement	Gastrostomy in drainage for 24 h Only intravenous hydration Monitoring of vital parameters (blood pressure, heart rate, diuresis, body temperature) and of the volume and quality of gastric drainage for 24 h, every 3 h Blood count test In the absence of complications, the administration of enteral nutrition within 24 h after the PEG positioning

**Table 2 nutrients-16-02893-t002:** Main socio-demographic, clinical, and nutritional characteristics of patients at the moment of PEG positioning at IRCCS INRCA Ancona vs. characteristics of patients with PEG treated with HEN.

	Patients at the Moment of PEG Positioning (*n* = 136)	Patients Treated with HEN (*n* = 88)
Gender; Absolute frequencies (%) *⸸	74 (54.41%) F; 62 (45.59%) M	56 (63.64%) F; 32 (36.64%) M
Age, mean ± SD ^	81.34 ± 9.19	81.90 ± 9.26
Reasons for hospitalization, Absolute frequencies (%) *	71 (52.17%) PEG positioning, 65 (47.83%) Other	50 (56.81%) PEG positioning, 38 (43.19%) Other
Albumin, Absolute frequencies (%) *	9 (6.82%) < 2.5 g/dL; 127 (93.18%) ≥ 2.5 g/dL	1 (1.14%) < 2.5 g/dL; 87 (98.86%) ≥ 2.5 g/dL
Diseases, Absolute frequencies (%) *		
Dementia ⸸	74 (54.14%)	54 (61.36%)
Other neuro	9 (13.43%)	34 (38.63%)
Cardio	20 (14.70%)	12 (13.63%)
DM2	22 (16.17%)	10 (11.36%)
Multimorbidity; Absolute frequencies (%) *	108 (79.41%) Yes; 28 (20.59%) No	68 (77.28%) Yes; 20 (22.72%) No
BMI, mean ± SD ^	20.90 ± 2.99	21.16 ± 2.95
Undernourished; Absolute frequencies (%) *⸸	75 (55.00%) Yes; 61 (45.00%) No	42 (47.73%) Yes; 46 (52.27%) No
Pressure Ulcers, Absolute frequencies (%) *⸸	50 (36.84%) Yes; 86 (63.16%) No	40 (45.45%) Yes; 48 (54.55%) No
Aspiration pneumonia; Absolute frequencies (%) *	15 (10.71%) Yes; 121 (89.29%) No	8 (9.09%) Yes; 80 (90.91%)
Length of hospital stay; Median (min, Max) ^	4 days (min 1; max 70)	3 days (min 1; max 70)

* Chi square; ^ t student, ⸸ *p* < 0.05.

**Table 3 nutrients-16-02893-t003:** The prevalence of HEN-related complications.

	Absolute Frequencies (%)
HEN-related complications	394 (100.00%)
PEG-related	278 (70.56%)
Tube-related	90 (22.84%)
Stomia-related	188 (47.71%)
Gastrointestinal	112 (28.43%)
Metabolic	4 (1.01%)

**Table 4 nutrients-16-02893-t004:** Major and minor PEG-related complications: absolute values, n. of complications/total PEG-related complications, n. complications/n. of patients.

	Complications	PEG-Related Complications/Complications	Patients with PEG-Related Complications/Tot Patients
Major complications			
Buried bumper	12	4.32%	5.88%
Minor complications			
PEG tube occlusion	6	2.16%	8.82%
PEG tube deterioration	22	7.91%	26.47%
PEG tube breakage	22	7.91%	14.70%
PEG tube dislocation	36	12.95%	23.53%
Granulation tissue	82	29.50%	29.41%
Peristomal site infection	84	30.21%	38.23%
Redness	2	0.72%	2.94%
Stoma enlargement	12	4.32%	14.71%

## Data Availability

The data presented in this study are available on request from the corresponding author. The data are not publicly available due to privacy restrictions.

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
