# Peer review of "Safety and Effectiveness of Percutaneous Endoscopic Gastrostomy May Be Improved by Proper Pre- and Post-Positioning Management of Elderly Patients with Multimorbidity"

_nutrients, 2024, doi:10.3390/nu16172893_

Round 1

Reviewer 1 Report

Comments and Suggestions for Authors

The authors present a retrospective analysis of a prospective observational study in 84 geriatric patients undergoing PEG. The data are of clinical interest and show clearly the benefits of a well established protocol and management. 

Diuscussion: The limitations of the study  should be added.

Line 87: A table for the PEG protocol may be considered

Line 91: For PEG programming the authors do not mention ascites - please explain 

Line 124: ...complications were extrapolated ?

Line 242 - 550.50%

Line 246 - The limitations of the study should be added

Comments on the Quality of English Language

final editing is recommended 

Author Response

We thank you for your suggestions, which were all very useful. We have accepted all the changes you proposed. Throughout the text we have highlighted in green the parts that have been added or modified. We also inform you that, following the introduction of two new bibliographical references (19,20), the numbers of the bibliographical notes have changed throughout the text.

We have used the manuscript version containing the revised text so that you can evaluate whether the improvements are satisfactory by comparing them with the original version.

  1. Discussion: The limitations of the study should be added.

We added the limitations in the discussion section (pg 8/9 of 11, highlighted in green)

  1. (ex) Line 87 – A table for the PEG protocol may be considered.

We added a table with all details for different phases (pre and post-positioning). (Table 1)

  1. (ex) Line 91: For PEG programming the authors do not mention ascites – please explain.

Thanks for pointing this out, it was an omission. We added also ascites. (Table 1, highlighted in green)

  1. (ex) Line 124… complications were extrapolated?

This was a poorly worded sentence that we changed at your suggestion. (pg 4 of 11, highlighted in green)

  1. (ex) Line 242 – 550.50%

Thanks for pointing out this error which we have corrected. (pg 6 of 11, highlighted in green)

  1. (ex) Line 246 : The limitations of the study should be added.
  2. We added the limitations in the discussion section (pg 8/9 of 11, highlighted in green)

Reviewer 2 Report

Comments and Suggestions for Authors

Congratulations for this research.

The article is interesting to read. However, several edits are required before publication can be considered:

1. Please rectify the last paragraph of the introduction to clearly specify the study hypothesis and objectives.

2. Include a PICO statement in the materials and methods section.

3. Perform statistical analysis in the results section between the study groups identified in the tables.

4. Include what statistical tests were used in the study.  Mention them in the statistical analysis section.

5. Table 1 must be split in 3-4 columns. Don't put all variables in the same column.

6. I would suggested creating at least 1 figure for the results.

7. Add a paragraph of study limitations in the discussion section.

8. Add a conclusion section

Best regards

Comments on the Quality of English Language

Minor edits required

Author Response

We thank you for your suggestions and observations which were all very useful. Throughout the text we have highlighted in green the parts that have been added or modified. We also inform you that, following the introduction of two new bibliographical references (19,20), the numbers of the bibliographical notes have changed throughout the text.

We have used the manuscript version containing the revised text so that you can evaluate whether the improvements are satisfactory by comparing them with the original version.

  1. Please rectify the last paragraph of the introduction to clearly specify the syudy hypothesis and objectives.

Following your suggestion, we modified the last paragraph of the introduction and specified the study hypothesis and objectives. (pg 2 of 11, highlighted in green)

  1.  
  2. Include a PICO statement in the materials and methods section.

Although it was a bit difficult and not exactly standard to use the PICO statement given the nature of our study, we added it as you suggested, together with two bibliographical references that also deal with the use of PICO in the context of descriptive studies. (pg 2 of 11, highlighted in green, references 19 and 20))

3 and 5. Perform statistical analysis in the results section between the study groups identified in the tables. Table 1 must be split in 3-4 columns. Don’t put all variables in the same column.

The answers to the two suggestions overlap in part. First, we have modified the table ex 1 now 2 and we hope we have interpreted your suggestion correctly. Furthermore, in this table, we have added information on the comparisons between the two groups and specified in the notes (as well as in the methodology) the tests used. As for other possible comparisons between the groups, the only possible comparison would have been relative to the data of the table ex 3 now

  1. We have performed these comparisons without finding any significance but, above all, we have doubts regarding the usefulness of such a comparison in consideration of the objectives of the study (that is why we didn’t report a requested data). The study aimed to describe the outcomes (complications and mortality) in a population subjected to the same management protocol pre and post PEG placement and investigate, as far as this was possible by a descriptive study, any advantages of a correct management pre and post PEG placement, comparing our data with those available in the literature.

  1. Include what statistical test were used in the study. Mention them in the statistical analysis section.

We added this information (pg 4/5 of 11)

  1. I would suggest creating at least 1 figure for the results.

Unfortunately, we have not understood exactly what kind of figure you suggest. We could make this further change after your explanations. Thanks

  1. Add a paragraph of study limitations in the discussion section.

Thank you for this suggestion, we added the study limitations (pg 8/9 of 11, highlighted in green)

  1. Add a conclusion section.

Thank you for this suggestion, we added conclusions (pg 9 of 11, highlighted in green)

Round 2

Reviewer 1 Report

Comments and Suggestions for Authors

Thank you for addressing the comments. 

Just one further suggestion; line 153 pluripathologies may be better multimorbid

Comments on the Quality of English Language

final editing is recommended

Author Response

Thank you for this suggestion!

We changed the term pluriphatology with multimorbid and multimorbidity in the text and in the table and highlighted it in red.

Best regards!

Reviewer 2 Report

Comments and Suggestions for Authors

Current version is significantly improved. Publication can be considered at this point. 

Comments on the Quality of English Language

English editing requires minor corrections.